Cloning and functional verification of Geraniol-10-Hydroxylase gene in Lonicera japonica

Zhang Shuping
Liu Zhenhua
Li Jia
Liu Qian
Zhang Yongqing
Pu Gaobin gbpu@163.com
College of Pharmacy, Shandong University of Traditional Chinese Medicine , Jinan, Shandong , China
Phitsuwan Paripok
Electronic publication date: 2025 Jan 13
Publication date: 2025
Volume: 13
Electronic Location ID: e18832
Received 2024 Mar 20; Accepted 2024 Dec 17
Copyright: © 2025 Zhang et al.
Copyright year: 2025
Copyright holder: Zhang et al.
License: This is an open access article distributed under the terms of the Creative Commons Attribution License, which permits unrestricted use, distribution, reproduction and adaptation in any medium and for any purpose provided that it is properly attributed. For attribution, the original author(s), title, publication source (PeerJ) and either DOI or URL of the article must be cited.
License URL: https://creativecommons.org/licenses/by/4.0/

Keywords: Lonicera japonica, Iridoid, Geraniol-10-hydroxylase, Overexpression, Silencing

Funding: Key Research and Development Program of Shandong Province 2022TZXD0036 Shandong Agricultural Seed Improvement Project 2021LZGC008 Chinese Herbal Medicine Innovation Team Project of Shandong Modern Agricultural Industry Technology System (SDAIT-20) This work was funded by the Key Research and Development Program of Shandong Province (Rural Revitalization Science and Technology Innovation Promotion Action Plan) (2022TZXD0036); Shandong Agricultural Seed Improvement Project (2021LZGC008); and Chinese Herbal Medicine Innovation Team Project of Shandong Modern Agricultural Industry Technology System (SDAIT-20). The funders had no role in study design, data collection and analysis, decision to publish, or preparation of the manuscript.

==============================
Background

Geraniol 10-hydroxylase (G10H) is a cytochrome P450 monooxygenase involved in regulation, which is involved in the biosynthesis of monoterpene. However, G10H is not characterized at the enzymatic mechanism and regulatory function in Lonicera japonica.

Methods and Results

A gene related to the biosynthesis of monoterpenoid, geraniol 10-hydroxylase, has been cloned from the medicinal plant Lonicera japonica. The gene, LjG10H, encodes a peptide of 498 amino acids with a predicted molecular weight of 54.45 kDa. LjG10H shares a homology of 72.93–83.90% with G10H from other plants. Phylogenetic analysis suggests that the protein encoded by this gene belongs to the cytochrome P450 monooxygenase family. Tissue-specific expression analysis revealed that LjG10H is most highly expressed in flowers. Through heterologous expression in E. coli, the LjG10H protein was purified and its catalytic activity was studied. The results show that the enzyme can catalyze the hydroxylation of geraniol to 10-hydroxygeraniol. Additionally, analysis of Lonicera japonica seedlings with silenced LjG10H revealed a reduction in monoterpenoid content.

Conclusions

This study demonstrates that LjG10H plays an important role in the biosynthetic pathway of iridoids. This is the first article that ascribes G10H to be associated with the biosynthetic pathway of iridoid. This study provides a theoretical basis for the functional mechanism of LjG10H in regulating iridoid synthesis and provides a valuable resource for molecular breeding studies.

Introduction

Honeysuckle (Lonicera japonica) is a plant in the Caprifoliaceae family, and its dried flower buds or flowers that are just beginning to bloom are commonly used as a large-volume medicinal material. Honeysuckle has the capacity to detoxify, disperse wind heat, and remove heat (Liu et al., 2020). So far, 420 secondary metabolites have been isolated from honeysuckle, including 87 flavonoids, 222 terpenoids, 51 organic acids, and other compounds (Ge et al., 2022). The urgent quest for new index components has garnered a lot of interest because of the uncertainty surrounding the interaction between luteolin and chlorogenic acid as well as the instability of its concentration. Iridoid, as one of the most diverse and high content components in Lonicera japonica, has become a breakthrough for new index. However, there are few reports on the synthesis pathways and regulatory mechanisms of iridoids in Lonicera japonica.

Monoterpenoids possess anti-inflammatory properties and are derived from oxygenated hemiacetals of ant odorous aldehyde. They are made up of two isoprene units (Hsu et al., 2016; Lv et al., 2021). These compounds are key components of Lonicera japonica and are frequently utilized in antimalarial and anti-influenza medications (Yuan et al., 2012; Huang et al., 2021). The biosynthetic pathways for monoterpenoids have been identified in several species (Zhang et al., 2022; Pickett, 2022). One significant rate-limiting step in the hydroxylation of geraniol is believed to occur at the C-10 site (Fig. 1). The conversion of geraniol to 10-hydroxygeraniol is being catalyzed by the enzyme geraniol 10-hydroxylase (G10H) (Hallahan & West, 1995). The vincristine content in Catharanthus roseus can be increased by overexpressing G10H (Pan et al., 2012), which was initially cloned from C. roseus in 2010 (Höfer et al., 2013). The activity of the G10H protein can be assessed, revealing its ability to catalyze the formation of 10-hydroxygeraniol when transferred into yeast cells (Collu et al., 2001). In Portulaca oleracea, this conclusion was further confirmed (Jeena, Kumar & Shukla, 2021). Subsequent research on the CrG10H gene in C. roseus demonstrated that CrG10H plays a crucial role in flavonoid synthesis by catalyzing the production of dihydronaringenin, which is derived from the 3′-hydroxylation of naringenin, in addition to geraniol (Sung et al., 2011; Huang et al., 2012; Wang et al., 2019). The full-length cDNA of SmG10H was cloned from Suaeda maritima by Wang et al. (2010a), and SmG10H was expressed heterologously in E. coli. It was discovered through in vitro study that SmG10H catalyzes the hydroxylation of geraniol. The transcript levels of SmG10H, along with the concentrations of sueda glycosides and 10-hydroxygeraniol increased dramatically in plants overexpressing SmG10H, suggesting that its involvement in the synthesis of cycloether terpenoids. Ophiorrhiza pumila produces camptothecin at a considerably higher rate than wild-type plants when the G10H and STR genes are co-expressed (Cui et al., 2015). The quantity of cornus alkaloids is elevated in the hairy roots of Cornus officinalis when G10H and ORCA3 are co-expressed (Wang et al., 2010b). Thus, the synthesis of alkaloids, monoterpenoids, and flavonoids is significantly regulated by the G10H gene.

Figure 1 Overview of metabolic pathways leading to the biosynthesis of iridoid.

The LjG10H gene was cloned, and the physicochemical properties of LjG10H were initially investigated in this study. LjG10H was produced and purified in vitro, and the catalytic activity of the purified protein molecules for geraniol was confirmed. It was established that LjG10H regulates the amount of monoterpenoids through in vivo transgenic studies and virus-induced gene silencing (VIGS) technology. This finding provides essential guidelines for the production and regulation of monoterpenoid compounds.

Materials and Methods

Materials

Plant materials

A total of 3-year-old Lonicera japonica plants grown in the Medicinal Plant Garden of Shandong University of Traditional Chinese Medicine were used for tissue-specific analysis. Seedlings obtained from harvested seeds grown in pots for 2 month were used for silencing expression analysis. The samples were washed, chopped, ground into powder under liquid nitrogen, and stored at −80 °C for later use. Three biological replicates of each treatment were performed.

Instrument materials

ExPASy (http://web.expasy.org/protparam/) was used to analyze the physicochemical properties of LjG10H encoded protein. The molecular formula and molecular structure of LjG10H protease were preliminarily studied. The phylogenetic tree of LjG10H protein was constructed by MEGA X, and the amino acid sequences of LjG10H and G10H from other seven different sources were compared by DNAMAN software.

The high-performance liquid chromatography (HPLC)

Chromatographic analysis was carried out using a Waters 2,695 high performance liquid chromatograph equipped with a quaternary pump, degasser, column heater (30 °C) containing a Zorbax Eclipse XDB-C18 column (250 × 4.6 mm i.d.; 5 μm particles) connected to a guard column. Peak identities were confirmed by comparing both with the retention times of the standards.

The mobile phase was a mixture of water (solvent A) and acetonitrile (solvent B). A gradient consisting of: 8–21% B in 30 min at 1 mL min −1 was used. The injection volume was 10 μL in all cases. The optimum detection wavelengths were 260 nm.

Cloning of the LjG10H gene

Based on the genomic sequencing results of Lonicera japonica, specific primers were designed (Table 1) for PCR amplification utilizing Lonicera japonica complementary DNA (cDNA) as a template. The amplification protocol was as follows: an initial denaturation at 95 °C for 3 min, followed by 34 cycles of denaturation at 95 °C for 30 s, annealing at 58 °C for 30, and extension at 72 °C for 1.5 min, concluding with a final extension at 72 °C for 5 min. The cloning products were analyzed using 1% agarose gel electrophoresis and subsequently purified with a gel recovery kit (TIANGEN, Beijing, China). The target fragments were ligated into a TA vector and transformed into E. coli (DH-5α; TaKaRa, Beijing, China). Positive colonies were selected, verified through colony PCR, and sequenced following overnight incubation at 37 °C on LB agar plates supplemented with kanamycin (Beijing Solarbio Science & Technology, Beijing, China).

Table 1 Primer sequences.

Primer name	Sequence (5′-3′)	Objective	
LjG10H	F: 5′-TATAGGATCCATGGATTTCTTCACCATTGCTC-3′	Gene cloning	
LjG10H	R: 5′-TATAAAGCTTTTACACTAGTGGACTCGGAAC-3′		
qRT-LjG10H	F: 5′- TTATTGCCGCAAGAGTAGCC-3′	qRT-PCR	
qRT-LjG10H	R: 5′- CTGGTACGGATCGGTCAAGT-3′		
Actin	F: 5′- CGTTGACTACGTCCCTGCCCTT-3′		
Actin	R: 5′- GTTCACCTACGAAACCTTGTTACGAC-3′		
pET-LjG10H	F: 5′-ACAAGGCCATGGCTGATATCGGATCCATGGATTTCTTCACCATTGCTCTC-3′	Prokaryotic expression	
pET-LjG10H	R: 5′-CAGTGGTGGTGGTGGTGGTGCTCGAGTTACACTAGTGGACTCGGAACAGC-3′		
CmLjG10H	F: 5′- TATAGGTACCGGAATATAATGGTGGAGGCTGG-3′	Silencing expression	
CmLjG10H	R: 5′- TATATCTAGATTGATGTTGTATCGGTCCCTG-3′		

Tissue-specific expression analysis of LjG10H

Total RNA was extracted from the roots, stems, leaves, flowers, and fruits of 3-year-old Lonicera japonica utilizing an RNA extraction kit (Vazyme, Nanjing, China). The integrity of the extracted RNA was assessed through electrophoresis on a 1% agarose gel, and the concentration was quantified using a Qubit fluorescence photometer and cDNA was synthesized employing a TaKaRa reverse transcription kit (TaKaRa, Beijing, China), and the cDNA was stored at −20 °C. Specific primers for qRT-PCR based on the cDNA of the LjG10H gene were designed (Table 1). The CFX96 Touch Real-Time PCR Detection System was utilized for the qRT-PCR analysis. The qRT-PCR reactions were conducted using a qRT-PCR kit (TaKaRa, Beijing, China) following the amplification protocol: 95 °C for 3 min; 95 °C for 10s, 54 °C for 30 s, and 72 °C for 1 min for a total of 40 cycles. Each sample was subjected to three technical replicates, and the relative expression of the LjG10H gene was analyzed using the 2−ΔΔCT method.

Prokaryotic expression of LjG10H

The prokaryotic expression vector pET32a was linearized through double digestion with Bam H I and Xho I (TaKaRa, Beijing, China) and subsequently ligated with the amplified LjG10H gene fragment, resulting in the construction of the plasmid pET32a-LjG10H. The recombinant plasmid was then transformed into competent E. coli BL21 (TaKaRa, Beijing, China), and positive colonies were verified via PCR and subsequently sent for sequencing. The plasmid confirmed by sequencing was extracted using a plasmid extraction kit (TIANGEN, Beijing, China) and transformed into E. coli BL21. Positive strains were cultured overnight and then inoculated into LB liquid medium containing ampicillin at a dilution ratio of 1:100. The culture was incubated at 37 °C with shaking at 200 rpm until the A600 reached between 0.6 and 0.8. Following this, 0.1 mmol/L isopropyl β-D-thiogalactoside (IPTG; TaKaRa, Beijing, China) was added, and the culture was induced at 16 °C for 16 h. Following induction, the bacteria were harvested via centrifugation at 5,000 rpm for a duration of 5 min. The bacterial pellet was then resuspended in 40 mL of cold equilibrium buffer, which was prepared by dissolving 1.64 g of sodium acetate and 29.25 g of sodium chloride in deionized water. To adjust the pH of the solution to 7.4, 1.17 mL of concentrated hydrochloric acid was added, and the final volume was adjusted to 1 L. The resuspended bacteria were subjected to sonication under the following parameters: power set at 200 W, sonication for 2.5 s, followed by a 5 s pause, repeated for a total of 80 cycles. Subsequently, the supernatant and precipitate were separated by centrifugation at 4,000 rpm and 4 °C for 5 min, after which both fractions were analyzed using SDS-PAGE. The supernatant was further purified utilizing a nickel agarose column chromatography technique.

Enzyme activity assay of LjG10H

To further assess the enzymatic activity of the purified protein, an in vitro reaction method was utilized to assess the enzyme activity of the recombinant protein. Geraniol served as the substrate, while the purified recombinant protein LjG10H acted as the catalyst in the enzymatic reaction. The reaction system comprised 1 IU of glucose-6-phosphate dehydrogenase, 4.5 mmol/L of glucose-6-phosphate (Shanghai Aladdin Biochemical Technology, Shanghai, China), 1 mmol/L of NADPH, and 20 μg of LjG10H protein. The incubation buffer consisted of a 50 mmol/L potassium phosphate buffer (pH 7.6) supplemented with 1 mmol/L EDTA, 1 mmol/L DTT, 10 mmol/L FAD, and 10 mmol/L FMN (Shanghai Aladdin Biochemical Technology, Shanghai, China). In this reaction, DTT serves to protect the reducing groups on the enzyme molecule and stabilize the activity of the enzyme. FMN functions as a crucial hydrogen and electron transporter within the respiratory chain, while FAD is primarily involved in the oxidative dehydrogenation of organic compounds, including fatty acids. Glucose-6-phosphate dehydrogenase catalyzes the conversion of glucose-6-phosphate and NADP+ into 6-phosphogluconic acid and NADP. Following a 5-min preincubation at 30 °C, geraniol (at a final concentration of 2.5 mmol/L) was introduced to initiate the reaction. The mixture was incubated for 3 h at 30 °C, after which methanol was added to terminate the reaction. Subsequently, the reaction mixture was centrifuged at 12,000 rpm for 5 min, and then analyzed using HPLC.

Analysis of the LjG10H gene silencing expression in Lonicera japonica

The pCMV201-2bN81 plasmid was double-digested with the restriction enzymes QuickCut Kpn I and QuickCut Xba I (TaKaRa, Beijing, China). The reaction system consisted of pCMV201-2bN81 plasmid 1 μg, QuickCut Kpn I 1 μL, QuickCut Xba I 1 μL, 10× QuickCut Buffer 5 μL, and ddH2O supplementation to 20 μL, and the enzyme was digested at 37 C for 30 min.

In order to construct the pCMV201-2bN81-LjG10H vector, the specific fragment of LjG10H, which had Kpn I and Xba I restriction sites, was amplified and primers LjG10H F and LjG10H R (Table 1) were designed using Snapgene. The PCR product was then linked to the digested pCMV201-2bN81 plasmid using T4 ligase (TaKaRa, Beijing, China), and transformed into E. coli (DH5). Recombinant plasmid pCMV201-2bN81-LjG10H with correct sequencing was taken to transform Agrobacterium receptive C58C1. The Agrobacterium tumefaciens C58C1 strain, harboring the plasmids pCMV101, pCMV201-2bN81, pCMV201-2bN81-LjG10H, and pCMV301, was cultured in LB medium supplemented with kanamycin at a concentration of 50 μg/mL and rifampicin at a concentration of 25 μg/mL. The culture conditions were maintained at 28 °C with shaking at 180 rpm for a duration of 48 h. The shaken bacterial solution at a 1:100 dilution was transferred into the 50 mL LB medium, to achieve an A600 value of Agrobacterium tumefaciens between 0.6 and 1.0. Following centrifugation at 4,000 rpm for 10 min, the bacterial cells were harvested, and the supernatant was discarded. An infection buffer was prepared by adding 10 mmol/L MgCl (Tianjin Dingshengxin Chemical Co., Ltd., Tianjin, China), 10 mmol/L MES (Shanghai Maclin Biochemical Technology Co., Ltd., Shanghai, China), and 100 μmol/L acetyl butyryl (AS, Shanghai Maclin Biochemical Technology Co., Ltd., Shanghai, China) to ddH2O to a final volume of 100 mL. The absorbance of the washed and resuspended bacterial cells was adjusted to A600 = 1.0, followed by incubation for a minimum of 3 h to induce protein expression. Subsequently, the bacteria harboring pCMV101, pCMV201-2bN81-LjLjG10H, and pCMV301 were mixed at a volume ratio of 1:1:1.

The healthy and robust Lonicera japonica seedlings were injected with a syringe. Using a needle to make a slight scar on the back of the cotyledon, a 1 mL sterile syringe without the needle was then used to align the wound and inject the mixed bacteria solution. The infected Lonicera japonica seedlings were first dark-treated for 48 h, and then cultured in a culture chamber (light 16 h/dark 8 h, 23 C, 60%ofhumidity). A total of 15 days later, new leaves were sampled to determine the relative expression level of LjG10H and iridoid content.

Determination of iridoid content

To quantify the iridoid content, the concentrations of loganic acid, morroniside, swertiamarin, loganin, and dehydrogenation loganin were detected in the sample. The leaves of transgenic Lonicera japonica were subjected to freezing and drying for a duration of 48 h, after which they were ground into a fine powder and passed through a sieve with a pore size of 4. A precise weight of 0.5 g was measured and placed into a 100 mL stoppered conical flask, followed by the addition of 50% methanol to achieve a final volume of 50 mL. The weight of the flask was recorded, and the mixture underwent ultrasound treatment for 1 h. Subsequently, the solution was allowed to stand at room temperature, and the weight was measured again to account for any loss, which was compensated by the addition of 50% methanol. The solution was then thoroughly mixed and centrifuged at 4,000 rpm for 10 min. Finally, the supernatant was filtered using a 0.22 μm organic filter membrane. The solution obtained was subjected to analysis via HPLC. The chromatographic conditions were established as follows: the mobile phase comprised 0.5% phosphoric acid (A) -acetonitrile (B), gradient elution: 0–5 min, 92% A; 5–35 min, 92–87% A; 35–43 min, 87–79% A; 43–48 min, 79% A; 48–60 min, 79–74%. The column temperature was 30 °C, the flow rate was set at 1 mL/min, and the sample volume was 10 μL. The detection wavelength was established at 240 nm.

Data processing

The relative expression level of the LjG10H gene in the leaves of Lonicera japonica was determined utilizing the 2−∆∆Ct method. Significant differences in gene expression were assessed using SPSS software, and a graphical representation was generated employing GraphPad Prism software.

Results

Cloning and sequence analysis of the LjG10H gene in Lonicera japonica

The LjG10H gene was successfully cloned from Lonicera japonica, and subsequent sequence analysis confirmed that the amplified full-length cDNA of LjG10H is 1,497 bp, encoding a protein of 498 amino acids. The LjG10H protein exhibited a homology range of 72.93–83.90% with seven other G10H amino acid sequences derived from various sources. The alignment of the protein sequences revealed that LjG10H possesses a conserved domain structure characteristic of most plant P450s, which includes a proline-rich region (PPGPxPLP) located at positions 33–40, a central helix (AGTDTT) at positions 302–307, and a heme-binding domain (PFGxGRRxCxG) at positions 432–442, consistent with the known characteristics of G10H in plants (Fig. 2B). Furthermore, a phylogenetic tree was constructed based on the alignment of 21 amino acid sequences, including LjG10H, which indicated that the LjG10H protein from Lonicera japonica is closely clustered with proteins from Valeriana jatamansi, suggesting a close genetic relationship (Fig. 2A).

Figure 2 Cloning and sequence analysis of LjG10H.

(A) Phylogenetic relationship between LjGl0H and G10H of different species, (B) amino acid sequence alignment of LjG10H with related proteins. (C) Expression of LjGl0H in different tissue, **p < 0.01.

Expression analysis of the LjG10H gene

The findings indicated that the expression of the LjG10H gene in flowers was significantly higher than in other tissues, with leaves exhibiting the next highest expression levels, while the expression in fruits was the lowest (Fig. 2C). This result aligns closely with previous research. Additionally, the concentration of iridoids was markedly greater in flowers compared to stems and leaves, where they were predominantly expressed. For example, the concentration of secoxyloganin in flowers was 34.4 times and 4.77 times greater than that in leaves and stems, respectively (Wang et al., 2023).

Prokaryotic expression of the LjG10H gene

Following induction with IPTG, a prominent band was observed at an approximate molecular mass of 72 kDa in both the supernatant and precipitate of the cells (Fig. 3, lane 5 and 6). This observation suggests effective expression, as the detected mass exceeded the expected value of 55.45 kDa, likely due to the presence of the thioredoxin (TRX)-tag. The protein was successfully purified through elution with 5% imidazole (Fig. 3, lane 7). Conversely, no bands were detected in Lane 2 when 30% imidazole was employed to elute the cleaved supernatant, which may be attributed to the high concentration of imidazole.

Figure 3 The 10% SDS-PAGE analysis of the expression and purification of recombinant LjG10H in E. coli BL21.

1: Maker; 2: imidazole at 30%; 3: 5% imidazole; 4: upper column effluent; 5: cell lysate supernatant; 6: cell lysis and precipitation; 7: purified protein.

Detection of the LjG10H enzyme activity

In the enzyme activity detection system, geraniol served as the substrate for HPLC analysis. The retention times for the standard 10-hydroxygeraniol and geraniol were recorded at 5.0 and 25.2 min, respectively (Figs. 4A and 4B). Within the reaction system designed for the purification of LjG10H protein, the retention time of the target product was observed at 5.1 min (Fig. 4C). The intensity of the target peak increased upon the introduction of 10-hydroxy geraniol standard into the same system, confirming its identification as 10-hydroxygeraniol (Fig. 4D). Conversely, in the reaction that did not include LjG10H protein, no peak was detected at this retention time (Fig. 4E). These results suggest that LjG10H exhibits catalytic activity in the conversion of geraniol to 10-hydroxygeraniol in vitro.

Figure 4 Enzymatic activity of recombinant LjG10H on pregnenolone.

(A) Geraniol standard; (B) 10-hydroxygeraniol standard; (C) geraniol and LjG10H incubation 3 h; (D) C and 10-hydroxygeraniol standard; (E) geraniol incubation 3 h.

Silencing plant screening and content determination

To investigate the specific role of LjG10H in Lonicera japonica, a silencing expression system was established, and five silenced expression Lonicera japonica seedlings were selected for quantitative reverse transcription polymerase chain reaction (qRT-PCR) analysis of their newly developed leaves. The results indicated that, in comparison to normally growing Lonicera japonica seedlings, the relative expression level of the LjG10H gene was significantly diminished (Fig. 5A). The expression levels for Line-1, Line-2, Line-3, Line-4, Line-5, and Line-6 were recorded as 0.26, 0.43, 0.50, 0.42, 0.13, and 0.32 times that of the wild type (WT), respectively. This finding suggests that the silenced expression plants were successfully generated, allowing for progression to subsequent experiments. Plants exhibiting a lower degree of gene silencing were subsequently selected for further validation to enhance the reliability of the gene silencing results. Among these, Lines 2, 3, and 4 demonstrated suboptimal levels of gene silencing efficacy and were therefore chosen for the determination of iridoid contents. The concentration of iridoid compounds in the newly developed leaves was analyzed using high-performance liquid chromatography (HPLC). In comparison to normally growing Lonicera japonica seedlings, the iridoid compound content in the silenced plants was significantly reduced to 0.58, 0.64, and 0.57 times that of the control group (Fig. 5B). These results indicate a positive correlation between the relative expression level of the LjG10H gene and the variation in iridoid compound content. The observed reduction in iridoid content in Lonicera japonica with a lower degree of silencing further corroborates that a more complete silencing leads to a more substantial decrease in iridoid content.

Figure 5 Silence ofpositive selection and content determination of plant.

(A) Silent LjG10H relative expression in the plant, (B) iridoid content in silenced Lonicera japonica plants. p < 0.05, **p < 0.01.

Correlation analysis

According to the Pearson correlation analysis, the correlation coefficient between iridoid content and the LjG10H gene was found to be 0.506. This correlation was statistically significant at the 0.01 level, indicating a strong and significant positive relationship between iridoid content and the LjG10H gene (see Table 2).

Table 2 Correlation analysis of iridoid content and LjG10H gene.

Relevance		LjG10H	Iridoids	
LjG10H	Pearson correlation	1	0.969**	
	Sig. Double tail		0	
	Number of cases	12	12	
Iridoids	Pearson correlation	0.969**	1	
	Sig. Double tail	0		
	Number of cases	12	12	
Note:

** At the 0.01 level (two-tailed), the correlation was significant.

Discussion

In recent years, G10H has been the subject of extensive research within various medicinal plants. A study conducted in 2001 investigated the specific role of G10H in periwinkle, indicating its involvement in the biosynthesis of terpenoid indole alkaloids. An overexpression analysis of the SmG10H gene in Swertia callus revealed elevated concentrations of 10-hydroxygeraniol and swertiamarin in comparison to the wild type, suggesting a specific regulatory effect of SmG10H on the synthesis of iridoid compounds. However, direct evidence supporting the involvement of LjG10H in the biosynthesis of iridoids in Lonicera japonica is still lacking. In the context of monoterpenoid biosynthesis, G10H serves as a crucial enzyme that facilitates the synthesis of intermediates (Dong et al., 2022). In this study, a G10H gene sequence was cloned from the genome of Lonicera japonica and designated as LjG10H. This gene exhibits a high degree of similarity to the VjG10H gene from Valeriana jatamansi. According to tissue-specific studies, LjG10H is expressed at significantly higher levels in the leaves and flowers of Lonicera japonica (Cai et al., 2019; Wang et al., 2023). The LjG10H gene appears to play a primary role in the production of monoterpenoids in these tissues, as evidenced by the increased concentration of monoterpenoids in honeysuckle flowers compared to stems. This study represents the first demonstration that the key enzyme LjG10H can catalyze the conversion of geraniol to 10-hydroxygeraniol through in vitro prokaryotic expression and silencing techniques, thereby confirming its significant role in regulating the content of monoterpenoids in Lonicera japonica.

The majority of contemporary in vivo research studies on G10H primarily focus on generation of stable transgenic callus or hairy roots. Notably, the concentration of terpenoid indole alkaloids was found to increase in the hairy root cultures of Catharanthus roseus upon the overexpression of G10H (Peebles et al., 2011; Pandey et al., 2016). Additionally, the SmG10H gene was successfully integrated into the hybrid callus tissue of Bupleurum chinense through somatic cell hybridization, resulting in the production of novel compounds, specifically swertiamarin and mangiferin. This study confirmed that the SmG10H gene exerts a significant regulatory influence on the accumulation of swertiamarin (Wang et al., 2011).

The successful silencing of the LjG10H gene in Lonicera japonica, and the subsequent confirmation of its role in regulating the synthesis of monoterpenoid compounds, represents a novel achievement in this species. However, due to the transient nature of virus-induced gene silencing (VIGS) technology, the extent of gene repression may eventually increase over time. Consequently, seedlings of Lonicera japonica were selected for this study. Quantitative reverse transcription polymerase chain reaction (qRT-PCR) was employed to further validate the effective silencing of LjG10H in the Lonicera japonica plants. Comparative analysis of the monoterpenoid content in silenced plants vs. that in normally growing Lonicera japonica seedlings, conducted through high-performance liquid chromatography (HPLC), indicated a significant reduction in both LjG10H expression and monoterpenoid levels. Additionally, the investigation revealed a negative correlation between G10H and gardenia glycoside concentrations in gardenia fruits, particularly as the fruits matured (Du et al., 2021). Furthermore, treatment of rehmannia with 5-azaC demonstrated a positive correlation between G10H and the accumulation of iridoid glycosides, as well as the expression levels of related enzyme genes (Xu et al., 2023). The findings presented are consistent with the silencing experiment conducted on LjG10H, thereby providing further evidence that LjG10H plays a role in the biosynthesis of cycloartane. However, it is important to note that LjG10H was not completely silenced due to technological limitations. Although there was a reduction in monoterpenoid content, it was not entirely eliminated. This situation necessitates further validation through stable genetic transformation. Following this, high-quality germplasm resources will be identified utilizing genetic markers associated with the LjG10H gene. This approach will lay a theoretical foundation for the development of novel germplasm resources of Lonicera japonica, as well as facilitate the selection and breeding of new varieties.

Conclusions

Our research suggests that G10H serves as an effective regulatory target for the metabolism of iridoid synthesis, particularly in Lonicera japonica. This study demonstrates for the first time that the overexpression of G10H is adequate to enhance iridoid synthesis in this species. Consequently, the overexpression of LjG10H in Lonicera japonica may represent a promising strategy for increasing iridoid yield in the near future.

Supplemental Information

Supplemental Information 1 Prokaryotic expression of the original SDS-PAGE map.

Supplemental Information 2 PCR of LjG10H.

Amplification using Lonicera japonica cDNA as a template.

Supplemental Information 3 MIQE Checklist.

Supplemental Information 4 Raw data.

qRT-PCR and HPLC results for tissue-specific expression and silencing method expression.

Additional Information and Declarations

Competing Interests

Author Contributions

Data Availability

The authors declare that they have no competing interests.

Shuping Zhang conceived and designed the experiments, performed the experiments, analyzed the data, prepared figures and/or tables, and approved the final draft.

Zhenhua Liu conceived and designed the experiments, authored or reviewed drafts of the article, and approved the final draft.

Jia Li conceived and designed the experiments, authored or reviewed drafts of the article, and approved the final draft.

Qian Liu conceived and designed the experiments, authored or reviewed drafts of the article, and approved the final draft.

Yongqing Zhang conceived and designed the experiments, authored or reviewed drafts of the article, and approved the final draft.

Gaobin Pu conceived and designed the experiments, authored or reviewed drafts of the article, and approved the final draft.

The following information was supplied regarding data availability:

The raw data is available in the Supplemental File.

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
