# Peer review of "Cloning and functional verification of Geraniol-10-Hydroxylase gene in Lonicera japonica"

_PeerJ, doi:10.7717/peerj.18832_

## Round 0.1 · original submission · Major Revisions

Please respond to all reviewers' comments, particularly in the Methodology and Discussion sections

Reviewer 1 ·

Basic reporting

1. The quality of the figure, especially the alphabet in Figure 1-3, needs to be improved to high resolution with more than 300 dpi. In Figure 2B, the first Proline (P) of proline-rich region was concealed with the red block, please edit to show the correct sequence (PPGPPLP).
2. In Figure 2A, what is the difference between the DNA sample in lane 1 and 2.
3. Please add more descriptive information in the figure legend throughout the manuscript. Please add the chemical’s name under chemical structure that would be beneficial for the reader to understand synthesis pathway.

Experimental design

1. Basically, G10H specifically catalyzes hydroxylation of geraniol to 10-hydroxygeraniol in a single step. Sung et al., 2011 (https://doi.org/10.1021/jf200259n) have shown that the reaction for enzyme assay contains substrate, buffer, NADPH, and protein of interest. Therefore, please the authors explain in more detail about the function of components in enzyme assay reaction, for example DTT, FAD, FMN, glucose-6-phosphate dehydrogenase and glucose-6-phosphate.
2. Due to LjG10H gene expression in Lonicera japonica is tissue-specific in which highest expression level was observed in flower, followed by leaf. Please give more information why the content of iridoids in transgenic Arabidopsis thaliana overexpressed with L. japonica LjG10H gene was measured from leaves. Why this experiment can represent the role of LjG10H related with monoterpenoids synthesis in different parts of plants.
3. Please add more detail in materials and methods section, for example:
1) Add the information related with sequence annotation and phylogenetic analysis methods, as well as bioinformatics tools
2) Add the information of E. coli strain used the for recombinant protein production
3) Add the information of HPLC column that used for geraniol and 10-hydroxygeraniol analysis, as well as type of detector

Validity of the findings

1. Regarding recombinant LjG10H expressed in E. coli under T7 promoter in pET32 plasmid (Figure 4), the target protein band (approximately 75 kD) larger than expected size (55.45 kDa) might be resulted from thioredoxin (TRX)-tag (12 kDa), not from His-tag (6 amino acids) as the authors mentioned in line 213.
2. Regarding LjG10H gene silencing expression in Lonicera japonica to study the specific role of LjG10H in the plant (Figure 8), only iridoid content of silenced plants line-2, line-3, and line-4 comparing with wild type was shown in Figure 8. However, lack of iridoid content of line-1, line-5, and line-6 with lower LjG10H gene expression. It would be beneficial for the reader if the authors add more data to show correlation between LjG10H gene expression level and iridoid content in wild type and silenced lines of Lonicera japonica. This concept would be also applied to Figure 6.

Reviewer 2 ·

Basic reporting

The manuscript reports isolation and functional characterization of geraniol 10-hydroxylase (G10H), a key enzyme in iridoid biosynthetic pathway, in Honeysuckle (lonicera japonica). The authors should address the following comments in the revised version.

Comments:

Introduction: some of the statements made in this section are confusing and difficult to follow. Please modify the sentence and make it clear. Also, please make sure the references correctly cited.

line 56-57 “The research has indicated that the transfer of G10H into C. roseus cells does not reveal any protease activity.”
line 69-70 Cui et al. 2015: Is it on Rauwolfia?

M&M: Arabidopsis results are not that important to this study.

line 153 loganin acid? loganic acid.

line 153-154 Brassicaceae family plants, with a few exceptions, do not accumulate iridoids. how did the authors measure/quantify these compounds in control? Is G10H overexpression sufficient to induce accumulation of these iridoids?

M&M section (line # ): VIGS: This is one of the important experiments in this study. It is not clear (i) how the construct was made, (ii) which vector was used for VIGS, (iii) how the experiment was performed, (iv) age of the seedling, (v) how silencing efficiency was determined. What is CmLjG10H F/R? VIGS is used for many plant species. I strongly suggest the authors take look at published literature. The entire section should be clearly written.

Results:
Figure 6C: It appears to me the wild type Arabidopsis accumulates iridoids. what are those iridoids and how was it quantified?

First of all, experiment on the native plant (L. japonica) should come before Arabidopsis. Arabidopsis transformation is not necessary here.
I want to see a clear description of VIGS. How were the metabolites quantified? Did the authors use any standard?
Assuming that the authors did it correctly, why did they choose lines # 2, 3, and 4 for metabolite analysis.
why not Line 1, 5, 6?
Figures:
Figure 2A: not necessary. Combine Figure 2 B, C with Figure 3
Figure 2: correct the scientific name: Rauwolfia serpentine
Figure 6A: not necessary
Figure 6B: this is a heterologous expression. how was relative expression measured here? Relative to what? Include a simple RT-PCR picture or use low expression line as a control to show the relative expression.

Experimental design

Please see my comments above.

Validity of the findings

Please see my comments above.

Additional comments

Please see my comments above.

---

## Round 0.2 · Major Revisions

Please kindly respond to the comments carefully:

- Clearly address and describe the methodology.
- Provide a detailed description of the source of materials used in the experiment.
- In the results section, I suggest the authors begin with the hypothesis, explaining why this experiment was conducted and the criteria for selecting the materials used in the study.
- Importantly, the discussion should be improved. It should relate to the results and compare them with the literature. A comprehensive discussion is required.
- Carefully check citations, figure captions, and numbering.
- Improve the overall English language quality.

Reviewer 1 ·

Basic reporting

no comment'

Experimental design

no comment'

Validity of the findings

1. The figure legends of figure 2 A, B, and C do not correlate with the figures. Regarding the picture below, A should be phylogenic relationship, B should be amino acid sequence alignment, and C should be expression of LjG10H. Therefore, please carefully revise that point both in the figure legend and main text (Line 215-230). In addition, please add the box to indicate the conserved domains in amino acid sequence alignment of P450 homologs.
2. What is the protein sample in Figure 3 lane 2, and what is reason of not getting any band in SDS PAGE.
3. Please check the species name throughout the manuscript, for example E. coli in the legend figure 3 need to be italic.

Additional comments

no comment'

Reviewer 2 ·

Basic reporting

The revised manuscript has number of issues.
There are mistakes/citation errors in the manuscript. The authors should have read the manuscript carefully before submitting to this journal.
The authors should seek help from someone proficient in scientific writing.

Comments:
For line numbers, see the manuscript with track changes.
Introduction:
line 56-67 “The research has indicated that the transfer of G10H into C. roseus cells does not reveal any protease activity” delete.
Collu et al. 2001: describes the cloning of G10H and biochemical characterization. describe correctly.
Cui et at. 2015: Is it related ajmalicine and Rauwolfia.
Wang et at: Is it related to Cornus officinalis?

M&M:
line 86 what was age of the seedlings used for VIGS.
line 92/93 LjG10H protein
line 96 HPLC
line 119 First strand cDNA was synthesized…
line 120 “concentration was 10 pmol/μl” concentration of primers?
line 133 E. coli: which E. coli strain was used for expression of recombinant protein?
line 140 pH 0.4?
line 150-161 description is lengthy, difficult to follow, not necessary. describe in few lines.
line 169-184 This section is poorly written and difficult to follow; should be rewritten correctly.
line 169/170 Agrobacterium tumetifolia or Agrobacterium tumefaciens?
line 185-186 pl. describe it correctly.
line 185-191 Vector construction should be described first.
line numbers: line 261 comes after line 191 what is missing here.
line 273 detected in sample: which sample?
line 274 leaves of transgenic Arabidopsis thaliana? why Arabidopsis?

Results
line 292 sequence analysis
line 299, 302: are these figure numbers correct?
line345 iridoid content: why line # 2, 3, and 4, the authors did not answer my question.

Discussion: I do not think the authors read and modify the Discussion section.
line 400 “in vivo overexpression in Arabidopsis”: Delete.
line 408-411 Delete, as Arabidopsis data is not there in the revised manuscript.
line 425 delete Arabidopsis here.

Experimental design

Please see my comments.

Validity of the findings

Please see my comments.

---

## Round 0.3 · Major Revisions

Please address the following concerns carefully, as expressed by one of the Section Editors:

"A) The writing needs to be polished; the content also has serious flaws. Starting from the very beginning (Abstract): "Background: Geraniol 10-hydroxylase (G10H) is one of the important regulatory cytochrome P450 monooxygenase, which is involved in the biosynthesis of monoterpene
alkaloids. However, G10H is not characterized at the enzymatic or at the regulatory aspect in Lonicera japonica."

1) should be "is a cytochrome P450 monooxygenase involved in regulation." Note: P450s are usually involved in biosynthesis and metabolism, not in regulation.
2) "monoterpene alkaloids"; Terpenes are not the same as alkaloids!!!! Geraniol is not an alkaloid.
3) " not characterized at the enzymatic or at the regulatory aspect": "regulatory aspect" is more a legal term. Better: "enzymatic mechanism and regulatory function."

The enzymatic reaction was only monitored with HPLC, and the shifts in retention times make the identification of compounds unreliable. Could the authors please add experimental proof for the identity of compounds, e.g., using an internal standard ("spiking") or LC-MS experiments?"

---

## Round 0.4 · Minor Revisions

- Please read the reviewers' comments carefully and make the necessary improvements to the manuscript.
- Abstract: For scientific clarity, geraniol 10-hydroxylase (G10H) is identified as a member of the cytochrome P450 monooxygenase family. G10H specifically catalyzes the hydroxylation of geraniol. Although the hydroxylation of geraniol is a rate-limiting step and G10H catalyzes this reaction, the regulatory functions of G10H, such as its enzyme kinetics and response to genetic perturbation through gene silencing, require further investigation.
- Figure 4: In the box, it should be labeled as "10-hydroxygeraniol."
- LC-MS analysis should be performed to identify the target molecules.

Reviewer 1 ·

Basic reporting

1. There are some grammatical errors that need to be corrected. In addition, please check the species name throughout the manuscript, for example E. coli in the legend figure 3 need to be italic.
2. Please correct the description of protein sample in SDS-PAGE from "Line" to "Lane" (Line 246-252).

Experimental design

-

Validity of the findings

-

Reviewer 2 ·

Basic reporting

I read the revised manuscript. I do not think the authors read their manuscript carefully. The authors should address or correct the following issues in the revised manuscript.

Comments:
Background: Geraniol 10-hydroxylase (G10H) is a cytochrome P450 monooxygenase which is involved in the biosynthesis of monoterpene. However, G10H is not functionally characterized in Lonicera japonica.

line 85 The phylogenetic tree of LjG10H protein
line 104 The target fragments were ligated into a TA vector
line 108-118 Delete this section (line 108-118), because the same information is repeated on the following page (line 132-142)
line 307-308 The majority of contemporary in vivo research studies on G10H primarily focus on generation of stable transgenic callus or hairy roots.

line 342 This study demonstrates for the first time that the overexpression of G10H is adequate to enhance iridoid synthesis in this species.

Experimental design

pl. see my comments.

Validity of the findings

no comments.

Additional comments

no comments.

---

## Round 0.5 · accepted · Accept

The manuscript is improved and can be accepted for publication. However, English could be further improved for clarity.